# The First Report of In Vitro Antifungal and Antibiofilm Photodynamic Activity of Tetra-Cationic Porphyrins Containing Pt(II) Complexes against *Candida albicans* for Onychomycosis Treatment

**DOI:** 10.3390/pharmaceutics15051511

**Published:** 2023-05-16

**Authors:** Ticiane da Rosa Pinheiro, Gabrielle Aguiar Dantas, Jean Lucas Gutknecht da Silva, Daniela Bitencourt Rosa Leal, Ricardo Barreto da Silva, Thiago Augusto de Lima Burgo, Roberto Christ Vianna Santos, Bernardo Almeida Iglesias

**Affiliations:** 1Graduate Program in Pharmaceutical Sciences, Center for Health Sciences, Department of Microbiology and Parasitology, Federal University of Santa Maria, Santa Maria 97105-900, Brazil; 2Laboratory of Experimental and Applied Immunology, Federal University of Santa Maria, Santa Maria 97105-900, Brazil; 3Department of Physics, Federal University of Santa Maria, Santa Maria 97105-900, Brazil; 4Bioinorganic and Porphyrin Materials Laboratory, Department of Chemistry, Federal University of Santa Maria, Santa Maria 97105-900, Brazil; 5Department of Chemistry and Environmental Sciences, Ibilce, São Paulo State University (Unesp), São Jose do Rio Preto 15054-000, Brazil

**Keywords:** photodynamic therapy, *Candida albicans*, onychomycosis, Pt(II) porphyrins, antibiofilm activity

## Abstract

Onychomycosis is a prevalent nail fungal infection, and *Candida albicans* is one of the most common microorganisms associated with it. One alternative therapy to the conventional treatment of onychomycosis is antimicrobial photoinactivation. This study aimed to evaluate for the first time the in vitro activity of cationic porphyrins with platinum(II) complexes **4PtTPyP** and **3PtTPyP** against *C. albicans*. The minimum inhibitory concentration of porphyrins and reactive oxygen species was evaluated by broth microdilution. The yeast eradication time was evaluated using a time-kill assay, and a checkerboard assay assessed the synergism in combination with commercial treatments. In vitro biofilm formation and destruction were observed using the crystal violet technique. The morphology of the samples was evaluated by atomic force microscopy, and the MTT technique was used to evaluate the cytotoxicity of the studied porphyrins in keratinocyte and fibroblast cell lines. The porphyrin **3PtTPyP** showed excellent in vitro antifungal activity against the tested *C. albicans* strains. After white-light irradiation, **3PtTPyP** eradicated fungal growth in 30 and 60 min. The possible mechanism of action was mixed by ROS generation, and the combined treatment with commercial drugs was indifferent. The **3PtTPyP** significantly reduced the preformed biofilm in vitro. Lastly, the atomic force microscopy showed cellular damage in the tested samples, and **3PtTPyP** did not show cytotoxicity against the tested cell lines. We conclude that **3PtTPyP** is an excellent photosensitizer with promising in vitro results against *C. albicans* strains.

## 1. Introduction

Onychomycosis is one of the most prevalent fungal infections among existing nail disorders [1,2]. Yeasts of the genus *Candida* are emerging pathogens associated with onychomycosis, especially in South America, Africa, and Asia, where the hot climate favors the proliferation of these microorganisms [3,4,5]. Species of the genus *Candida*, such as *Candida albicans*, are natural constituents of the human microbiota that behave as opportunistic pathogens in the presence of homeostatic changes in the body, causing infections [6,7,8,9]. In addition, they can form a biofilm, producing an extracellular polymeric matrix that firmly adheres to the solid surface, protecting the microbial population [10,11,12,13,14,15]. The usual treatments for onychomycosis are restricted to oral and topical antifungal agents. Topical administration has the disadvantage of low penetration into the nail bed where the microorganism develops, requiring extended periods of use and the possible recurrence of the infection. Oral antifungals penetrate the nail bed and plate, although the commonly reported side effects make adherence to treatment difficult [16,17,18,19]. 

Antimicrobial photoinactivation is an alternative therapy that combines visible light, a photosensitizing compound (PS), and molecular oxygen. Soon after being irradiated, the PS absorbs light and can trigger photooxidation, generating radical species or singlet oxygen, cytotoxic products capable of causing oxidative damage to lipids, proteins, and DNA, leading to cell death [20,21,22]. The use of *meso*-tetra-substituted porphyrins such as PS has shown excellent results since inserting loads modulates PS activity, which interacts better with cellular biomolecules, improving the photoinactivation efficiency [23,24,25].

The discovery of new antifungal drugs is moving slowly while fungal resistance is increasing; in this context, antimicrobial photoinactivation is a possible alternative therapy in the fight against bacterial/fungal infections because, in addition to being a non-invasive technique, evidence has shown that it has resistance mechanisms and can be used in comorbid individuals [1,26,27]. Thus, this study sought to evaluate, for the first time, the in vitro antifungal and antibiofilm photodynamic activity of tetra-cationic porphyrins with peripheral Pt(II) complexes against onychomycosis *C. albicans*.

## 2. Materials and Methods

### 2.1. Samples and Culture Conditions

Four clinical isolates of *C. albicans* from onychomycosis were provided by the Mycological Research Laboratory of the Federal University of Santa Maria (UFSM), and three standard American-Type Culture Collection (ATCC) strains *C. albicans* ATCC 24433, *C. albicans* ATCC 14053, and *C. albicans* ATCC SC5314 were provided by the Oral Microbiology Research Laboratory of UFSM. This study was approved by the Federal University of Santa Maria Research Ethics Committee (CAAE no. 65409222.0.0000.5346).

### 2.2. Photosensitizers

Four tetra-cationic porphyrins were used (Figure 1). Two porphyrins containing Pt(II) complexes **3PtTPyP** and **4PtTPyP** were previously synthesized according to the method by Naue et al. [28] and fully characterized according to the method by Tasso et al. [29]. As a comparative study, water-soluble tetra-cationic methylated porphyrins **3MeTPyP** and **4MeTPyP** were used and purchased from Frontier Scientific^®^ (Logan, UT, USA). The platinum(II) porphyrins tested were soluble in DMSO and stable in this solution.

### 2.3. Light Source

The photoinactivation assays were performed under white-light LED irradiation (400–800 nm, visible range) with an irradiance at 50 mW/cm^2^ and a light dosage at 360 J/cm^2^ for 120 min. According to the standard protocol, the irradiated 96-well plates were kept closed (10 cm from the light) [30]. The experiments were performed in triplicate. 

### 2.4. Minimum Inhibitory and Fungicidal Concentrations

The minimum inhibitory concentrations (MIC) of the porphyrins and antifungal agents ciclopirox olamine and fluconazole (Sigma-Aldrich^®^, St. Louis, MO, USA) were determined by the broth microdilution technique, according to the M27-A3 protocol of the Clinical Laboratory Standards Institute, with modifications [31]. In summary, 100 µL of Sabouraud broth (KASVI^®^) was added to the wells of a microplate, and later, in column 1, 100 µL of the tested treatment (porphyrin or antifungal) was added. From this, a serial dilution was performed for porphyrins (concentration range of 30–0.11 µM), ciclopirox olamine (concentration range of 241–15 µM), and fluconazole (concentration range of 836–13 µM), except in column 11, which received only culture medium (negative control) and column 12, which received culture medium and inoculum (positive control). The inoculums were prepared in sterile 0.9% saline solution and standardized on a McFarland 1 scale in a densitometer (DEN-1^®^), followed by a 1:50 and 1:20 dilution in Sabouraud broth. After preparation, 100 µL of each inoculum (1:20) was added to the plate wells except for the negative control. The plates were incubated in two ways: one plate received white-light irradiation for 120 min with subsequent incubation of 24 h/37 °C, and the other plate was incubated in the dark for 24 h/37 °C. The MIC value was the lowest concentration capable of visually inhibiting fungal growth. 

The minimum fungicidal concentrations (MFC) were determined by seeding 1.0 µL aliquots from each well without visible fungal growth and positive and negative controls on Sabouraud dextrose agar plates (KASVI^®^) and incubating them for 24 h/37 °C. The MFC was defined as the lowest concentration without fungal growth on the plate. 

### 2.5. Cell Viability Curve Test

A cell viability curve assay was performed to determine the likely time that the treatment inhibited *C. albicans* growth. For this, the MIC of the porphyrin was previously determined against the isolates by the broth microdilution technique, as described above. Two 96-well plates were prepared using the following concentrations: 2×MIC, MIC, and ½MIC, in addition to the positive growth and negative controls. Next, one plate was exposed to white-light irradiation for 120 min at 37 °C, and the other plate was incubated at 37 °C in the dark. Aliquots of 10 μL were removed from the wells at different times (0, 10, 15, 30, 60, 120, and 180 min) and seeded on Sabouraud dextrose agar plates with a Drigalski loop. Plates were incubated for 24 h/37 °C. Subsequently, the colonies were manually counted, and the colony-forming units (CFU) were calculated and expressed in Log^10^CFU/mL [32].

### 2.6. ROS Scavenger Assay

To determine the presence of reactive oxygen species (ROS) in the photoinactivation process, five substances described in the literature as “scavengers” of ROS were used, namely ascorbic acid (AA), which detects the presence of singlet oxygen, N-acetylcysteine (NAC), which detects hydroperoxyl radicals, ethylene-diamine-tetra-acetic acid (EDTA), which is a metal ion chelator, *terc*-butanol (*t*-BuOH), which detects hydroxyl radicals, and dimethyl sulfoxide (DMSO), which detects superoxide radical species. The MIC of the isolated substances was determined by the microdilution assay in broth according to the protocol in Section 2.5. Then, a serial dilution of the porphyrins was performed in a 96-well plate, which was followed by adding 10 μL of the sequestering substances at fixed concentrations (10×½MIC) to the wells. Lastly, 100 μL of inoculum was added to the wells except for the negative control that only received the culture medium. Plates were irradiated with white light for 120 min and incubated for 24 h/37 °C. The ROS was determined when the MIC of the tested porphyrin increased in the presence of a sequestering substance. This method is well-established and easy to reproduce for identifying the photochemical mechanism, although it does not determine the location of the substance in the microbial cell [33].

### 2.7. Checkerboard Assay

The interaction of the porphyrins with the antifungals was tested using the checkerboard technique according to the method by Vitale and collaborators [34]. In a separate experiment, a series of two-fold dilutions of each treatment (antifungal and porphyrin) in the corresponding solvents were performed using microtubes. Aliquots (50 µL) of each porphyrin concentration were added to columns 2 through 11, and 50 µL aliquots of each antifungal concentration were then added to rows B through H in a 96-well plate. In the wells of column 1 and the wells of row A, 50 μL of the culture medium was added. Thus, column 1 and row A only received the antifungal and porphyrin, respectively, and were used as control wells to determine the MICs of the isolated compounds. Column 12 was split into positive and negative controls. Lastly, 100 μL of the standardized inoculum was seeded in each well except for the negative control. The plate was irradiated with white light for 120 min and then incubated for 24 h/37 °C. The porphyrin–antifungal interaction analysis was calculated by the fractional inhibitory concentration index (FICI) using the following equation: FICI = MIC AB/MIC A + MIC BA/MIC B, where MIC AB and MIC BA are the respective inhibitory concentrations of the compounds in association, and these are divided by MIC A and MIC B (the inhibitory concentrations of the isolated compounds). The interaction was classified as synergistic (FICI ≤ 0.5), indifferent (0.5 < FICI < 4.0), or antagonistic (FICI ≥ 4.0). 

### 2.8. In Vitro Biofilm Formation and Destruction

*C. albicans* biofilms were formed according to Krom et al. [35] and Vila et al. [36], with some modifications. The *C. albicans* isolates were previously cultivated in Sabouraud dextrose agar for 24 h/37 °C. Then, yeasts (2 to 3 isolated colonies) were transferred to 5 mL of trypticase soy broth (TSB) (Laborclin^®^, Caxias do Sul, Brazil) and incubated for 24 h/37 °C under constant agitation at 150 rpm. Afterwards, the culture was diluted (1:100) in TSB supplemented with 2.0% glucose (TSB 2.0%). The inoculum was homogenized by vortexing, and 200 μL was added to the wells of a 96-well microplate, except for the negative control wells that received only TSB 2.0%. The plate was incubated for 90 min at 37 °C under constant agitation at 150 rpm so the yeasts could adhere to the surface of the plate (adhesion phase). The suspensions were carefully aspirated, and 200 µL of fresh 2.0% TSB was added to the wells. The plate was again incubated for biofilm maturation for 48 h/37 °C under constant agitation at 150 rpm. Biomass quantification in the formed biofilm was performed using the 1.0% crystal violet technique. The supernatant was aspirated, the wells containing biofilm were washed with sterile 0.9% saline solution three times to remove non-adherent cells, and the plate was dried in an oven at 60 °C for 60 min. After fixation, the biofilms were stained with 200 µL of crystal violet 1.0% for 20 min, followed by three washes with sterile 0.9% saline solution to remove excess dye. In the end, 200 µL of 95% ethanol was added for 10 min to elute the crystal violet and transferred to a new plate where the absorbance reading was performed in a microplate reader (Bio-Rad 550^®^) at 570 nm [37]. 

Then, to evaluate the ability of the porphyrins to reduce the preformed biofilm, the MIC and two higher concentrations (2×MIC and 4×MIC) were used. The positive and negative biofilm controls received no treatment. The plate was irradiated with white light and incubated for 24 h/37 °C. After incubation, the biofilm biomass was quantified using the crystal violet 1.0% technique detailed above. 

### 2.9. Atomic Force Microscopy

Atomic force microscopy (AFM) was performed in an NX10 instrument (Park Systems, Suwon, Republic of Korea) equipped with the SmartScan^®^ software (version 1.0) RTM 11a. Topography and adhesion maps were acquired simultaneously using the PinPoint nanomechanical mode. The AFM maps were obtained using a high-frequency rotated monolithic silicon probe (TAP300-G Budget Sensors, Bulgaria) with a nominal resonance frequency of 300 kHz and a 40 N/m force constant. All measurements were made under ambient conditions at a room temperature of 21 ± 5 °C and a relative humidity of 50 ± 10%. Images were treated offline using the XEI software (version 4.3.4,build 22, RTM1) [38,39,40]. 

### 2.10. Cytotoxicity

A cytotoxicity assay was performed according to the method by Mosmann et al. [40] using the MTT (3-[4,5-dimethylthiazol-2-yl]-2,5-diphenyltetrazolium) reduction technique. The cells used were human keratinocytes (HaCaT) and murine fibroblast cells (L929) purchased from the Rio de Janeiro Cell Bank. The cells were cultured in Dulbecco’s modified eagle medium with a low glucose content, supplemented with 10% fetal bovine serum and penicillin/streptomycin (100 U/L), and incubated at 37 °C in a humidified atmosphere containing 5.0% CO_2_. Soon after, the cells were seeded in 96-well plates, with 3.0 × 10^4^ cells/well for L929 and 4.0 × 10^4^ cells/well for HaCaT. The plates were pre-incubated for 24 h at 37 °C. As soon as the treatment with different concentrations of porphyrins (4×MIC, 2×MIC, and MIC) was added, the plates were irradiated for 120 min (irradiance of 50 mW/cm^2^ and total light dosage of 360 J/cm^2^) and incubated again for 24 h/37 °C. The MTT reagent was added, and the plates were incubated for 4 h/37 °C. The supernatant was aspirated, and DMSO was added to the wells to dissolve the formed formazan crystals. The absorbance was read in a microplate reader at 570 nm. 

### 2.11. Statistical Analysis

Statistical analysis was performed using one-way analysis of variance (ANOVA) followed by Tukey’s test. Data were represented as mean ± standard deviation. The graph Prism 8.01 software was used, and results were considered significant when *p* < 0.001. 

## 3. Results and Discussion

### 3.1. General Considerations

The main advantages of antimicrobial photodynamic therapy (aPDT), a promising alternative, include reduced side effects to the host, a minimally invasive method, a broad microbial spectrum, and an efficacy independent of antimicrobial resistance, unlike the antifungals used. Additionally, the development of a resistance mechanism is unlikely because the photodynamic process is a non-selective treatment capable of simultaneously damaging multiple targets [41,42]. Tetra-cationic porphyrins with peripheral Pt(II) complexes are compounds that generate ROS when activated with adequate light irradiation, including the radical species of the type-I mechanism (electron transfer) and mainly singlet oxygen (cell cytotoxic product) through the type-II mechanism (energy transfer) [29,43]. In addition, they are promising PSs of great interest in PDT, as they have adequate photophysical, photobiological, and photochemical properties in the application of photodynamic processes, are considered stable in DMSO solution (the effect of the solvent on cell viability was discarded) and buffers, they are photostable when irradiated with white LED light, and they do not tend to form aggregates [44,45,46].

### 3.2. MIC and MFC Analysis

The MIC and MFC values of the **3PtTPyP**, **4PtTPyP**, **3MeTPyP**, and **4MeTPyP** against the clinical isolates and standard *C. albicans* strains in the dark and under white-light irradiation are listed in Table 1. In these assays, the meta isomeric porphyrin **3PtPyP** showed the best results for MIC and MFC, demonstrating excellent in vitro antifungal activity when irradiated with white light (irradiance of 50 mW/cm^2^ and total light dosage of 360 J/cm^2^ for 120 min), resulting in the photoinactivation of the yeasts at low concentrations of the compound. Although the mechanism of photoinactivation against *C. albicans* is not fully elucidated, studies with different cationic porphyrin PSs have indicated, through fluorescent markers, that porphyrin binds to the yeast cell wall, and only after irradiation was the presence of the PS in the cytosol of the yeast observed in the cell, suggesting that the photodynamic process favors high cell permeability, causing irreversible damage [47,48]. The excellent in vitro antimicrobial activity of **3PtTPyP** has been reported against mycobacteria, dermatophyte fungi, and viruses [30,41,49]. Photodynamic studies against *C. albicans* using porphyrin *meso*-tetra[4-(3-N,N-dimethylaminopropoxy)phenyl] (TAPP) and its derivatives (TAPP^4+^), *meso*-tetra-(4-N-methylpyridyl) cationic porphyrins (TMPyP), and *meso*-tetra(4-sulfonated) anionic porphyrins (TPPS^4−^) have been found, although this is the first study to evaluate the activity of **3PtTPyP** against these yeasts [50,51,52,53]. Considering the best MIC and MFC values obtained, the **3PtTPyP** was the PS of choice for the other photoinactivation tests.

The porphyrins **4PtTPyP**, **3MeTPyP**, and **4MeTPyP** were less efficient in terms of photoinactivation, presenting low cytotoxicity to the fungal cells (Table 1), with little or no difference in photoinactivation in the dark or white-light irradiation cycles. The MIC and MFC values under white-light conditions were not satisfactory, as the necessary dose of the compound available for the photoinactivation process was high.

The values found for the **3MeTPyP** and **4MeTPyP** porphyrins may be related to the higher solubility of the compound in an aqueous solution, given that these porphyrins were diluted with water, which would hinder the interaction and permeability in the complex cell wall of fungi [22]. Although the **4PtTPyP** porphyrin is practically equivalent to **3PtTPyP** in terms of ROS generation, the Pt(II) complex coordinated to the N-pyridyl position in the molecule’s structure makes it slightly less soluble in water, DMSO, buffer solution, and others. It can form aggregates, decreasing its bioavailability for the photoinactivation process and corroborating the results obtained from the MIC and MFC values in this study [46]. The MIC value for the antifungal ciclopirox olamine was 15 µM for all the *C. albicans* strains tested in this study, and the MIC for the antifungal fluconazole was 52 µM for all the clinical isolates tested and *C. albicans* SC5314, while for *C. albicans* ATCC 14053, the MIC was 13 µM. The MIC remained the same under white-light irradiation and in dark conditions, proving that the antimicrobial agents tested were not photosensitive.

### 3.3. Cell Viability Curve Test

The cell viability curve assay was performed to determine the kinetics of the in vitro fungicidal activity of **3PtPyP** against two standard strains of *C. albicans*. The results were expressed by the viable cell count on a logarithmic scale compared with the positive growth control (without contact with the treatment) under white-light irradiation and in dark conditions (Figure 2 and Figure 3). After treatment with the selected porphyrin at the previously defined concentrations, we observed that photoinactivation of the *C. albicans* ATCC 14053 strain occurred within the first 60 min of exposure to a white-light source, causing the complete eradication of the tested microorganism (Figure 2a), and as expected, the concentration of ½MIC did not inhibit or reduce the CFU/mL^−1^, the samples remaining with viable cells even after 180 min. The test also showed that the presence of porphyrin in dark conditions led to the complete inhibition of fungal growth only at the highest concentration of **3PtTPyP** tested (30 µM) and in the longest time tested (180 min). Furthermore, it did not demonstrate antifungal activity at MIC and ½MIC (Figure 2b).

The standard strain *C. albicans* ATCC SC5314 was also tested to assess cell viability after porphyrin treatment (Figure 3). The photoinactivation process for this strain occurred in the first 30 min of exposure to a white-light source (Figure 3a), leading to fungal eradication at low PS concentrations (0.45 and 0.22 µM). In dark conditions, the result was similar to that obtained for the *C. albicans* ATCC 14053, in which only the concentration of 2×MIC (30 µM) could inhibit the microorganism growth (Figure 3b).

The results clearly showed that the inactivation was dependent on the irradiation source, and with the association of a PS, light and molecular oxygen could inhibit the fungal growth quickly. Basso and collaborators tested the action of **3PtTPyP** against enveloped viruses and reported the complete inactivation of the microorganism at low PS concentrations within 30 and 60 min when irradiated [49]. In addition, Rossi and coworkers described the bactericidal action of porphyrin **3PtTPyP** against mycobacteria in the first 24 h after irradiation [38].

### 3.4. ROS Scavenger Assay

ROS determination was carried out to evaluate which sequestering substances were part of the photooxidation process. In this way, each substance is directly related to ROS generation. These reactive species can damage proteins, lipids, nucleic acids, and induce structural changes that lead to a loss of cell function, cell apoptosis, or fungal virulence inhibition [50,51,52,53,54].

Given the results of the MIC values obtained in the absence and the presence of ROS scavengers shown (Table 2), we concluded that the mechanism involved in the photooxidation of *C. albicans* tested with the **3PtTPyP** porphyrin was mixed (type I + type II), that is, the type-I mechanism with the increased MIC in the presence of NAC (hydroperoxyl scavenger species—•OOH), showing hydroperoxyl radical production, and the type-II mechanism with the increased MIC in the presence of AA (singlet oxygen scavenger species—^1^O_2_), revealing singlet oxygen production, formed by the energy transfer process, to be a phototoxic product of the fungal cells.

The tetra-cationic porphyrins containing the peripheral platinum(II) complexes tested in this study are PSs that considerably absorb light from the visible spectrum and, in the presence of molecular oxygen, become more bioavailable, producing ROS, thus ensuring photophysical properties. Furthermore, they slightly tend to form aggregates and are stable in solution, making them a target of photochemical interest as a possible alternative in treating cutaneous and subcutaneous infections. Its applications in photooxidative processes have already been reported elsewhere [29,43,55]. Other studies have shown a mechanism of action involving singlet oxygen production for **3PtTPyP** and other cationic porphyrins against bacteria, mycobacteria, and dermatophytes [22,45,49,56]. Singlet oxygen species are the main photoinactivation products of tetra-cationic porphyrins, especially those of Pt(II) bipyridyl complexes. They are highly reactive and oxidizing species that act on cellular targets such as the cell wall, outer membrane, and intracellular components such as DNA [28,57].

### 3.5. Checkerboard Assay

The MIC values obtained for the porphyrin **3PtPyP** in the presence of ciclopirox olamine and fluconazole drugs are shown in Table 3. For the *C. albicans* ATCC SC5314 strain, the combination of the compounds (porphyrin + drug) reduced the MIC values obtained separately for each compound. However, this reduction was not enough to generate a synergistic interaction effect, keeping the effect indifferent in the presence of the other. The strain *C. albicans* ATCC 14053 showed a different behavior in the association of the compounds, as only the MIC of the antifungal in the presence of **3PtTPyP** decreased. In contrast, the MIC of the **3PtTPyP** in the presence of antifungals remained the same as the isolated compound. Nevertheless, the interaction was indifferent.

The association of antifungal compounds versus porphyrin using PDT against yeasts is insufficient to establish the best combination of agents. The number of studies is limited, and the techniques used are diverse. This is in addition to the specificities of each microorganism, making direct comparisons difficult. However, the synergistic effect of fluconazole and *meso*-tetra(N-methyl-4-pyridyl)porphyrin tetratosylate (TMP) and the indifferent effect of miconazole with TMP against yeasts has already been described elsewhere [58,59].

The indifferent effect reported in some studies corroborates our findings, although a synergistic effect of combined therapy in vitro has not been proven. The additive effect suggests that the associated use of antifungal versus photodynamic therapy does not generate an antagonistic effect, which is undesirable.

### 3.6. In Vitro Biofilm Formation and Destruction

The formation of the *C. albicans* biofilm occurs in a complex and orchestrated way, requiring controlled steps to develop. The main step in forming the *C. albicans* biofilm is the adhesion to the surface phase, where physicochemical interactions and the initial formation of microcolonies occur due to agitation. The biofilm is formed as soon as the cells begin to produce an extracellular polymeric matrix (maturation), which takes 48 to 72 h. To develop a biofilm formation, different culture media (Sabouraud broth, TSB, Mueller Hinton broth) supplemented with different concentrations of glucose (0, 1.0, and 2.0%) were tested for three incubation times (24, 48, and 72 h) in a static form and under stirring at 150 rpm. After the analyses, the best culture medium, supplementation, time, and conditions were TSB 2.0% glucose for 48 h under agitation. To evaluate the capacity of the porphyrin **3PtTPyP** to destroy the preformed biofilm, we used 4×MIC, 2×MIC, and MIC values. The results can be seen in Figure 4.

Given the results obtained after the biomass quantification by the crystal violet technique, it was evident that the 3PtTPyP porphyrin could significantly reduce the preformed biofilm compared to the positive control. For the strain *C. albicans* ATCC 14053 (Figure 4a), the formed biofilm was reduced by 94.3% using 4×MIC, 89.7% when using to 2×MIC, and by 88.9% at MIC. As for *C. albicans* ATCC SC5314 (Figure 4b), the treatment reduced the biofilm by 83.7% when using 4×MIC, 77.9% when using to 2×MIC, and reduced the biofilm by 79.1% at MIC. Although the mechanism of the destruction of the PDT biofilm with porphyrins has not yet been entirely described, studies using photoinactivation with other non-porphyrin PSs suggest that the antibiofilm action can occur in different ways by binding the PS activated by light to the biofilm matrix and thus generating ROS, causing oxidative damage, as well as by destabilizing this matrix, increasing the permeability of the PS to the intracellular environment, causing damage to cytoplasmic components [60,61]. Corroborating our findings, Vila et al. developed a biofilm of *C. albicans* and *Fusarium oxysporum* on a nail model using aluminum and yttrium laser therapy and pulsed light and observed a reduction of approximately 60% of the biofilm for C albicans and 92% for *Fusarium oxysporum* [34]. In addition, an assay performed with cationic porphyrins affected the preformed biofilm of *Escherichia coli*, Staphylococcus aureus, and *C. albicans* strains [62]. Although in vitro assays cannot predict the representation of an antibiofilm response in vivo, we consider it a relevant tool for initial tests.

### 3.7. Atomic Force Microscopy Study

Representative 2D and 3D AFM maps for *C. albicans* ATCC 14053 under different treatments (with white-light conditions) are shown in Figure 5. Even under white-light exposure, the AFM maps displayed the common morphology of the *C. albicans* yeast strain [63], with the surface composed of relatively uniform packed cells with some protrusions (Figure 5a). In this case, the unicellular budding yeast had a spheroid shape with a diameter in the 2.0 to 3.0 μm range, which was also observed in previous AFM results [64]. The **3PtTPyP** porphyrin treatment (for both ½MIC and MIC) completely inhibited the *C. albicans* growth (Figure 5b,c), and no cell was found after many AFM runs. Although the unicellular budding yeast could be seen under optical microscopes at a ½MIC concentration, the sample preparation for the AFM map acquisition was a much more inhospitable environment for the *C. albicans* yeast strain growth. Moreover, the results from the adhesion force maps (Figure 6) showed that the porphyrin treatment increased the adhesive forces between the tip and the sample surface. The adhesion forces for all the treatments are shown in Table 4. Under white-light conditions, the average value of the surface adhesion force was 0.358 μN, increasing with the **3PtTPyP** porphyrin treatment to 0.381 and 0.510 μN for the ½MIC and MIC concentrations, respectively.

As seen in the AFM images, the *C. albicans* yeast strain samples underwent remarkable change after the **3PtTPyP** porphyrin treatment, with their growth being completely inhibited or the cells being destroyed. We recall that porphyrin treatments are highly effective against many microorganisms, such as bacteria strains [36], although most the AFM images only revealed bacteria deformation and/or cells growth reduction. In addition, nanomechanical imaging is a powerful tool to elucidate the adhesive properties of *C. albicans* [65,66]. Force–distance curves have been used to obtain the adhesion forces between bacterium–fungus pairs [67] and even visualize the main steps of the *C. albicans*–macrophage interaction [68,69]. Here, we showed that, although the *C. albicans* yeast strain was absent from the sample surface after the treatment, the adhesive forces were increased so that the residual mass was stuck to the surface, which could help to prevent future cell growth in this area (Table 4).

### 3.8. Cytotoxicity

The porphyrin **3PtTPyP** was diluted in DMSO and tested at three different concentrations defined from the highest MIC obtained among the tested yeasts, that is, the inhibitory concentrations obtained for the *C. albicans* strain ATCC 14053 (4×MIC—3.75 µM, 2×MIC—1.8 µM, and MIC—0.9 µM). Hydrogen peroxide (H_2_O_2_) was used as a positive control of cell death against HaCaT and L929 cells. After white-light irradiation for 120 min (50 mW/cm^2^ and total light dosage of 180 J/cm^2^), we observed that the DMSO compound and the H_2_O_2_ control significantly reduced the cell viability of the HaCaT cells compared to the positive growth control; however, the different porphyrin concentrations tested increased the cell viability of the cells not showing cytotoxicity (Figure 7a). In the L929 cells, only the cell death control (H_2_O_2_) effectively reduced the cell viability. Although the DMSO reduced the viability, it did not cause significant damage compared to the positive growth control (Figure 7b). Furthermore, the tested porphyrin concentrations stimulated cell proliferation.

Vizzotto et al. evaluated the cytotoxicity of a tetra-ruthenated cationic porphyrin (**H_2_RuTPyP**), where, after white-light irradiation, the PS caused cell death in a melanoma cell line without reducing the viability of the HaCaT cells tested, which was in line with our study, evidencing the promising use for porphyrins (e.g., PS) [70]. In 2022, Urquhart et al. showed that the PS they tested, in addition to not showing cytotoxicity, stimulated the viability of HaCaT and L929 cells, which was in line with our findings [71]. With this, we can state that the porphyrin **3PtTPyP**, in addition to being an excellent PS against yeasts, also did not damage human cells in vitro, making it a compound of great interest to be studied as an alternative therapy against infections.

## 4. Conclusions

For the first time, this study evaluated the in vitro antifungal and antibiofilm activity of tetra-cationic porphyrins containing peripheral Pt(II) complexes against *C. albicans* aimed at treating onychomycosis. We observed that a low concentration of the porphyrin **3PtTPyP** showed high antifungal potential when irradiated against *C. albicans* yeasts. The compound could eradicate yeast growth after a short period of irradiation. The possible mechanism of cellular phototoxicity was mixed with hydroperoxyl radical and singlet oxygen species production (a mixed mechanism). The association of **3PtTPyP** with antifungals showed indifferent activity under white-light irradiation. Preformed biofilms were significantly reduced when treated with different **3PtTPyP** concentrations. Atomic force microscopy revealed morphological damage in the *C. albicans* strains after white-light irradiation, and the compound did not show cytotoxicity against the tested HaCaT and L929 cells. Therefore, tetra-cationic porphyrins containing Pt(II) complexes are excellent photosensitizers and promising compounds in aPDT against superficial fungal infections.

## Figures and Tables

**Figure 1 pharmaceutics-15-01511-f001:**
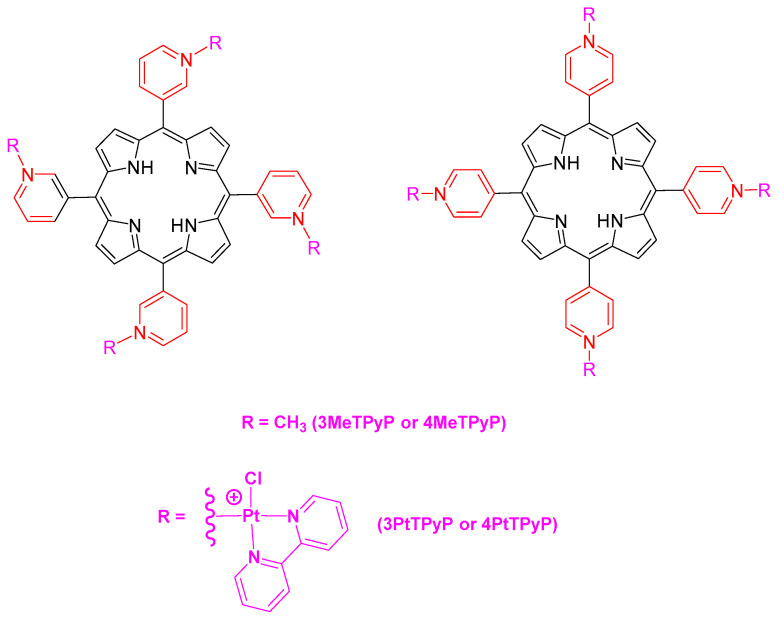
Structural representation of the studied tetra-cationic porphyrins **3MeTPyP**, **4MeTPyP**, **3PtTPyP**, and **4PtTPyP**. The counter-ions iodide (I^−^) and hexafluorophosphate (PF_6_^−^) are omitted from the figure for more clarity.

**Figure 2 pharmaceutics-15-01511-f002:**
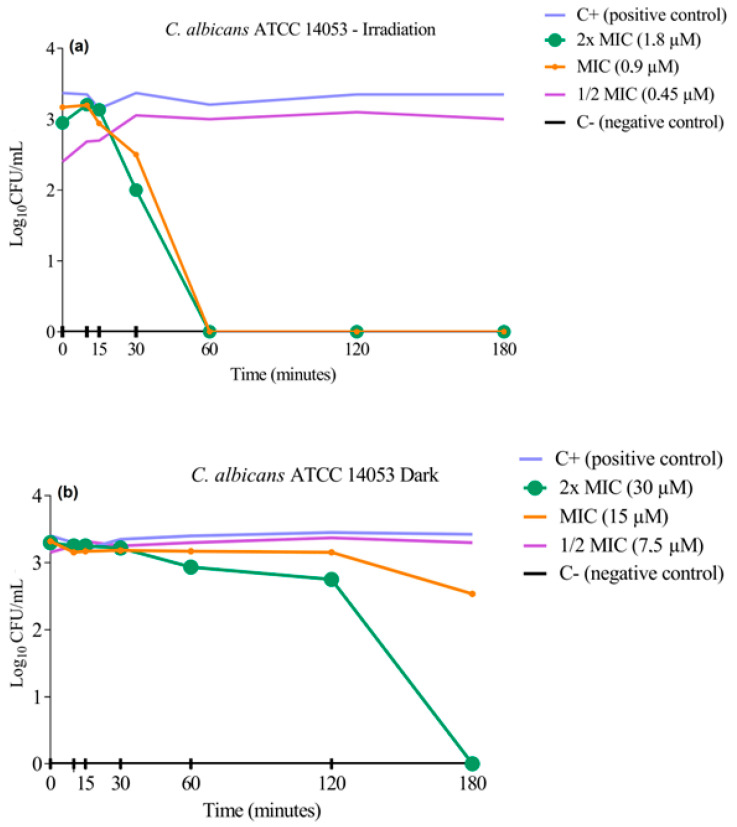
Cell viability curve of **3PtTPyP** against *C. albicans* ATCC 14053 (**a**) under white-light irradiation (50 mW/cm^2^ and total light dosage of 540 J/cm^2^) and (**b**) in dark conditions for a total time of 180 min.

**Figure 3 pharmaceutics-15-01511-f003:**
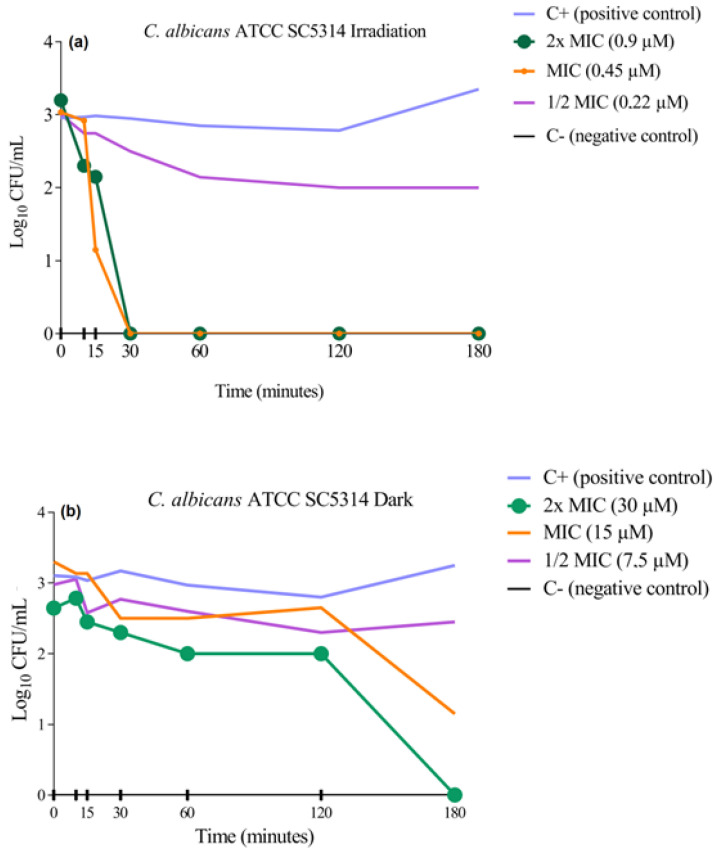
Cell viability curve of **3PtTPyP** against *C. albicans* ATCC SC5314 (**a**) under white-light irradiation (50 mW/cm^2^ and total light dosage of 540 J/cm^2^) and (**b**) in dark conditions for a total time of 180 min.

**Figure 4 pharmaceutics-15-01511-f004:**
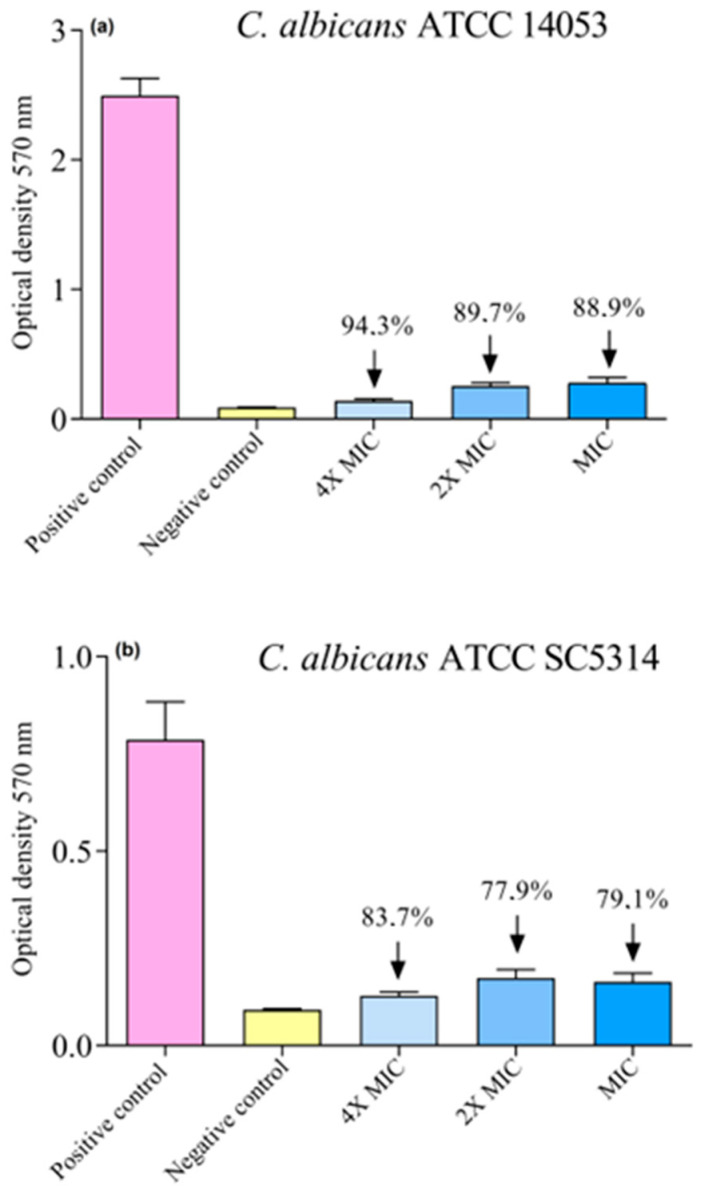
Biofilm formation and destruction of (**a**) *C. albicans* ATCC 14053 and (**b**) *C. albicans* ATCC SC5314 after treatment with **3PtTPyP** under white-light irradiation (50 mW/cm^2^ and total light dosage of 360 J/cm^2^) for 120 min.

**Figure 5 pharmaceutics-15-01511-f005:**
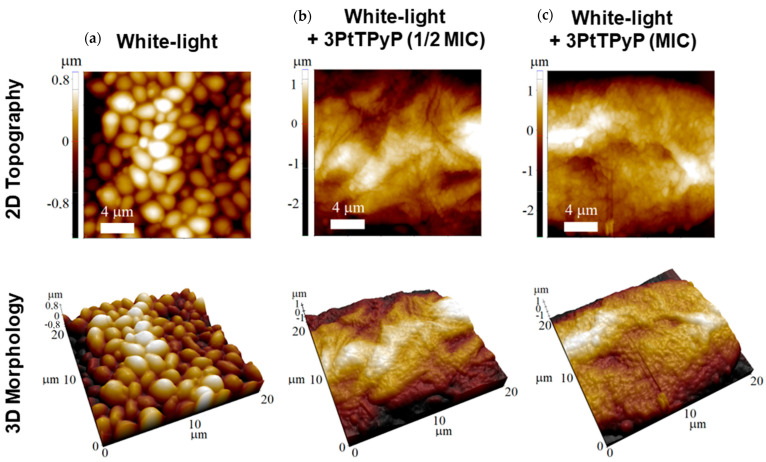
Atomic force microscopy topography images of *C. albicans* ATCC 14053 exposed to different protocols of photodynamic therapy using **3PtTPyP** porphyrin with an irradiance of 50 mW/cm^2^ and a total light dosage of 360 J/cm^2^ for 120 min.

**Figure 6 pharmaceutics-15-01511-f006:**
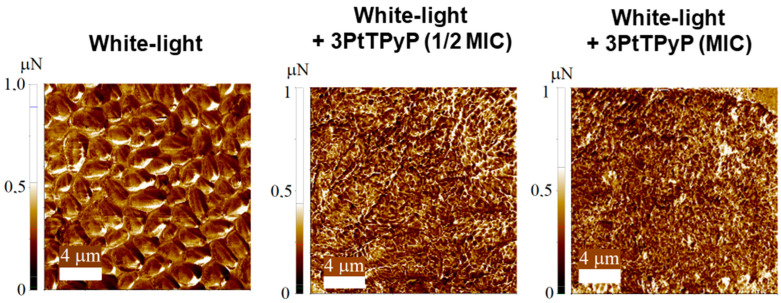
Adhesion force maps of *C. albicans* ATCC 14053 exposed to different protocols of photodynamic therapy using **3PtTPyP** porphyrin with an irradiance of 50 mW/cm^2^ and a total light dosage of 360 J/cm^2^ for 120 min.

**Figure 7 pharmaceutics-15-01511-f007:**
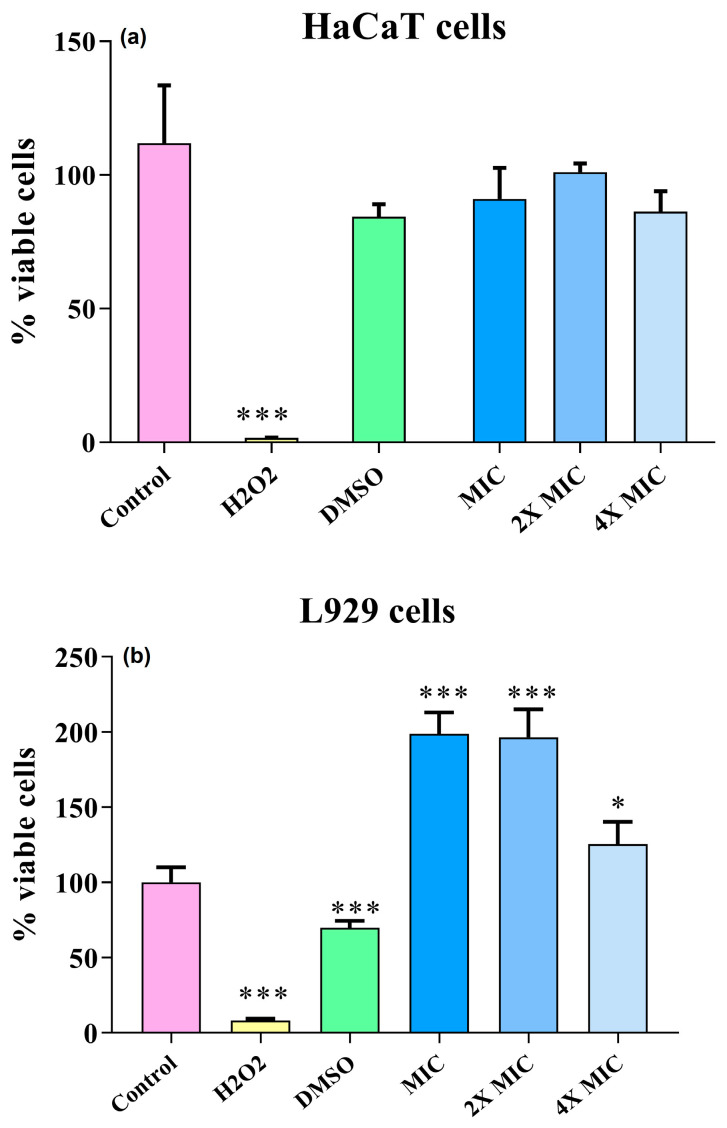
Cell viability of (**a**) HaCaT and (**b**) L929 cells using under irradiation for 120 min (50 mW/cm^2^ and total light dosage at 360 J/cm^2^). Values are expressed as mean ± standard deviation considering statistical difference when *p* < 0.001 compared with the control of viable cells (100%). Control (viable cells); H_2_O_2_ (negative control/non-viable cells); *** *p*< 0.001; * *p*< 0.05.

**Table 1 pharmaceutics-15-01511-t001:** MIC and MFC values (in μM) of **3PtPyP** and **4PtPyP** against *C. albicans* exposed to dark and white-light irradiation conditions of 50 mW/cm^2^ and total light dosage of 360 J/cm^2^ (120 min).

**Microorganism**	**3PtTPyP**			
	**MIC (µM)**		**MFC (µM)**	
	**Dark**	**Light**	**Dark**	**Light**
*C. albicans* CI ^1^ 03	15.0	0.11	15.0	0.22
*C. albicans* CI ^1^ 44	15.0	0.11	15.0	0.11
*C. albicans* CI ^1^ 45	15.0	0.90	15.0	0.90
*C. albicans* CI ^1^ 49	15.0	0.45	15.0	0.90
*C. albicans* ATCC 24433	15.0	0.45	30.0	0.45
*C. albicans* ATCC SC5314	15.0	0.45	30.0	0.45
*C. albicans* ATCC 14053	15.0	0.90	30.0	0.90
**Microorganism**	**4PtTPyP**			
	**MIC (µM)**		**MFC (µM)**	
	**Dark**	**Light**	**Dark**	**Light**
*C. albicans* CI ^1^ 03	15.0	7.50	15.0	15.0
*C. albicans* CI ^1^ 44	15.0	15.0	15.0	15.0
*C. albicans* CI ^1^ 45	15.0	15.0	30.0	15.0
*C. albicans* CI ^1^ 49	15.0	7.50	15.0	7.50
*C. albicans* ATCC 24433	15.0	15.0	15.0	15.0
*C. albicans* ATCC SC5314	15.0	7.50	15.0	7.50
*C. albicans* ATCC 14053	7.50	7.50	15.0	15.0

^1^ Clinical isolate.

**Table 2 pharmaceutics-15-01511-t002:** MIC values (μM) of microorganisms tested for the porphyrin **3PtTPyP** in the absence and presence of ROS scavengers under white-light irradiation conditions (50 mW/cm^2^ and total light dosage of 360 J/cm^2^) at 120 min.

Microorganism	3PtTPyP					
	MIC ^2^ (µM)					
	Absence	AA ^3^	NAC ^4^	EDTA ^5^	*t*-BuOH ^6^	DMSO ^7^
*C. albicans* CI ^1^ 03	0.11	7.50	7.50	0.05	0.11	0.11
*C. albicans* CI ^1^ 44	0.11	7.50	7.50	0.11	0.11	0.11
*C. albicans* CI ^1^ 45	0.90	7.50	7.50	0.45	0.45	0.90
*C. albicans* CI ^1^149	0.45	7.50	7.50	0.11	0.45	0.45
*C. albicans* ATCC 24433	0.45	7.50	7.50	0.22	0.22	0.45
*C. albicans* ATCC SC5314	0.45	7.50	7.50	0.22	0.45	0.45
*C. albicans* ATCC 14053	0.90	7.50	7.50	0.45	0.45	0.90

^1^ Clinical isolate; ^2^ minimum inhibitory concentration; ^3^ ascorbic acid; ^4^ N-acetylcysteine; ^5^ ethylenediamine tetra-acetic acid; ^6^
*terc*-butanol; ^7^ dimethyl sulfoxide.

**Table 3 pharmaceutics-15-01511-t003:** MIC values (μM) of porphyrin **3PtTPyP** in interaction with ciclopirox olamine and fluconazole against standard strains of *C. albicans* under white-light irradiation conditions (50 mW/cm^2^ and total light dosage of 360 J/cm^2^) at 120 min.

**Microorganism**	**3PtTPyP**					
	**3PtTPyP (A)**	**CO ^1^** **(B)**	**MIC ^2^** **A/B**	**MIC** **B/A**	**FICI ^3^**	**Int. ^4^**
*C. albicans* ATCC SC5314	0.45	15.0	0.22	1.56	1.0	I ^5^
*C. albicans* ATCC 14053	0.90	15.0	0.90	1.56	1.5	I
**Microorganism**	**3PtTPyP (A)**	**FLU ^1^** **(B)**	**MIC ^2^** **A/B**	**MIC** **B/A**	**FICI ^3^**	**Int. ^4^**
*C. albicans* ATCC SC5314	0.45	52.0	0.22	8.0	1.0	I ^5^
*C. albicans* ATCC 14053	0.90	13.0	0.90	2.0	1.5	I

^1^ Ciclopirox olamine or fluconazole; ^2^ minimum inhibitory concentration; ^3^ fractional inhibitory concentration index; ^4^ interaction; ^5^ indifferent.

**Table 4 pharmaceutics-15-01511-t004:** Adhesion force values of *C. albicans* ATCC 14053 exposed to different protocols of photodynamic therapy using **3PtTPyP** porphyrin with an irradiance at 50 mW/cm^2^ and a total light dosage of 360 J/cm^2^ for 120 min.

Conditions	Adhesion Force (µN)
Whitelight	0.358 ± 0.018
Whitelight + **3PtTPyP** (½MIC)	0.381 ± 0.050
Whitelight + **3PtTPyP** (MIC)	0.510 ± 0.069

## Data Availability

All analyzed data are contained in the main text of the article. Raw data are available from the authors upon request.

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
