# Peer review of "The First Report of In Vitro Antifungal and Antibiofilm Photodynamic Activity of Tetra-Cationic Porphyrins Containing Pt(II) Complexes against Candida albicans for Onychomycosis Treatment"

_pharmaceutics, 2023, doi:10.3390/pharmaceutics15051511_

Round 1
Reviewer 1 Report
The manuscript deals with an actual aim of antifungal photodynamic therapy. Two of the photosensitizers used by the authors are widely used in this field, namely tetracationic tetrapyridylporphyrins. The other two are not typical and the obtained results are of basic interest in the field of porphyrin photodynamic applications.
The manuscript could be published after some revision and clarification.
Comments and remarks:
1. Incorrect dimensions of values are used along the manuscript, such as mW/cm-2 and similar. It should be either mW/cm2 or mW.cm-2. The same with other values.
2. “Tetraplatinated porphyrin” is an incorrect term.
3. In general, the language of the manuscript requires corrections.
4. In the case of platinum-containing complexes the structure shown at fig. 1 is not correct. In contrast to methyl-substituted derivatives, N-Pt bond is coordinational, not covalent. In this respect, the main positive charge remains at metal ion, not the nitrogen.
5. The tetrametyl-substituted tetrapyridylporphyrins contain iodide counter-ions which could form iodine under irradiation. How do authors prove, that this process does not occur during the experiment?
6. At fig. 2a a rising of the ½ MIC curve is observed. How can authors explain such behavior?
7. In my opinion, in biological studies the purity of the compounds under investigation should be supported by the complete spectral characterization even if they are purchased from commercial suppliers. And must be supported if the compounds are synthesized by the authors. In this respect I insist, that the spectral data, particularly NMR and UV-vis, should be provided as supporting information.
Author Response
Reviewer#1
The manuscript deals with an actual aim of antifungal photodynamic therapy. Two of the photosensitizers used by the authors are widely used in this field, namely tetra cationic tetrapyridylporphyrins. The other two are not typical and the obtained results are of basic interest in the field of porphyrin photodynamic applications. The manuscript could be published after some revision and clarification.
Comments and remarks:
- Incorrect dimensions of values are used along the manuscript, such as mW/cm-2 and similar. It should be either mW/cm2 or mW.cm-2. The same with other values.
Response: Thank you. Corrected.
- “Tetraplatinated porphyrin” is an incorrect term.
Response: Thank you. Corrected.
- In general, the language of the manuscript requires corrections.
Response: Thanks for the comment. The manuscript was corrected by a specialized person and the certificate is attached in the submission of the revised work.
- In the case of platinum-containing complexes the structure shown at fig. 1 is not correct. In contrast to methyl-substituted derivatives, N-Pt bond is coordinational, not covalent. In this respect, the main positive charge remains at metal ion, not the nitrogen.
Response: Thank you. Corrected.
- The tetrametyl-substituted tetrapyridylporphyrins contain iodide counter-ions which could form iodine under irradiation. How do authors prove, that this process does not occur during the experiment?
Response: Thanks for the comment. Tetra-methylated porphyrins were purchased commercially. They have already been used in other photoinactivation studies and at no time has this iodine formation been verified. The presence of iodine can be verified by the starch test. In our last published articles (please see references 70 and 72), tests were conducted with potassium iodide and in no case did this formation occur and because of this, the reviewers asked to remove it from the manuscript.
- At fig. 2a a rising of the ½ MIC curve is observed. How can authors explain such behavior?
Response: We thank the reviewer. The increase in fungal growth observed in the first 0, 15 and 30 minutes in the ½ MIC curve probably represents the exponential phase of the microorganism that after a few minutes reached the stationary phase and remained until the end of the test.
- In my opinion, in biological studies the purity of the compounds under investigation should be supported by the complete spectral characterization even if they are purchased from commercial suppliers. And must be supported if the compounds are synthesized by the authors. In this respect I insist, that the spectral data, particularly NMR and UV-vis, should be provided as supporting information.
Response: Thanks for the comment. Tetra-methylated porphyrins were purchased commercially. Porphyrins containing platinum(II) complexes were synthesized on a large scale (close to 1g) and all data regarding purity and structure confirmation are present in the supplementary material of reference 43.

Reviewer 2 Report
Dear Authors,
the presented manuscript is well-design and focused on the topic of photodynamic inactivation in the treatment of onychomycosis. Authors selected clinical isolates for initial testing but then they used only standard strain. I would like to know why. Figure 4 showed only one standard strain with the ability to form biofilm and subsequent photoinhibition, it can be interesting to compare this result with other strains as well as with different photosensitizers. The same for the other following experiments.
Line 152 -there is mentioned "item 4.4" Where is this part, please? This experiment with ROS scavengers assay was a suitable choice but the authors must describe better the used protocol.
Mainly, there are missing supplementary data mentioned by the authors.
Author Response
Reviewer #2
- The presented manuscript is well-design and focused on the topic of photodynamic inactivation in the treatment of onychomycosis. Authors selected clinical isolates for initial testing but then they used only standard strain. I would like to know why.
Response: The authors thank the reviewer. The normal flow of work in our laboratory starts with clinical isolates (from different anatomical sites and different susceptibility profiles) to expand as much as possible the possibilities of studying the spectrum of action of prophyrins. In addition, we thought of ATCC strains to facilitate the future reproduction of results by other research groups.
- Figure 4 showed only one standard strain with the ability to form biofilm and subsequent photoinhibition, it can be interesting to compare this result with other strains as well as with different photosensitizers. The same for the other following experiments.
Response: We thank the reviewer. Although the optical density observed in the C. albicans ATCC SC5314 strain was lower than the optical density observed in C. albicans ATCC 14053, both strains formed biofilms when compared to the negative control. The reviewer presents an excellent suggestion. But this is a future possibility, because currently we do not have some materials in our laboratory and the import process could take up to a year (due to new Brazilian customs rules). In addition, due to the global situation due to the coronavirus pandemic, logistical and financial issues are extremely complicated. The development agencies in our country are at a critical moment and research funding is collapsing.
- Line 152 -there is mentioned "item 4.4" Where is this part, please? This experiment with ROS scavengers assay was a suitable choice but the authors must describe better the used protocol.
Response: We are sorry for the error mentioned on line 152, the correct one would be "item 2.5". Therefore, in item 2.5 we describe the principle of the method used and later in item 2.7 the aliquots used in the development of the method were reported.
- Mainly, there are missing supplementary data mentioned by the authors.
Response: Thanks for the comment. As already answered to reviewer#1, tetra-methylated porphyrins were purchased commercially. Porphyrins containing platinum(II) complexes were synthesized on a large scale (close to 1g) and all data regarding purity and structure confirmation are present in the supplementary material of reference 43.
Thank you for your kind consideration,
Yours sincerely,

Round 2
Reviewer 2 Report
Dear Authors,
thank you for your responses, and corrections and I understand your explanation regarding testing all included strains in this study.